# Supplemental UV-A Affects Growth and Antioxidants of Chinese Kale Baby-Leaves in Artificial Light Plant Factory

Rui He, Meifang Gao [ID], Yamin Li [ID], Yiting Zhang, Shiwei Song [ID], Wei Su and Houcheng Liu *

College of Horticulture, South China Agricultural University, Guangzhou 510642, China; ruihe@stu.scau.edu.cn (R.H.); gmf@stu.scau.edu.cn (M.G.); yaminli@stu.scau.edu.cn (Y.L.); yitingzhang@scau.edu.cn (Y.Z.); swsong@scau.edu.cn (S.S.); susan_l@scau.edu.cn (W.S.)
* Correspondence: liuhch@scau.edu.cn; Tel.: +86-020-85-280

**Abstract:** To investigate the effects of supplementary UV-A intensity on growth and antioxidant compounds in Chinese kale (*Brassica alboglabra* Bailey) baby-leaves, three different UV-A intensity treatments (5, 10, 15 W·m$^{-2}$, respectively) were applied 10 days before harvest in artificial light plant factory. In Chinese kale baby-leaves, supplemental 5 and 10 W·m$^{-2}$ UV-A (UVA-5 and UVA-10) were beneficial for inter-node length, stem diameter, canopy diameter, fresh weight and dry weight, particularly in UVA-10 treatment, while these above-mentioned growth parameters all significantly decreased in UVA-15 treatment. The soluble sugar content decreased under UVA-5, but there was no significant difference under UVA-10 and UVA-15. Soluble protein contents decreased under UVA-5 and UVA-10, but significantly increased under UVA-15. UVA-10 played a predominant role in increasing FRAP and contents of total phenolics and total flavonoids compared to other treatments. Contents of total glucosinolates (GLs), aliphatic GLs and indolic GLs in Chinese kale baby-leaves significantly increased with UV-A intensity increasing, and the highest contents were found under UVA-15. The percentage of total aliphatic GLs (about 80%) was significantly higher than those of total indolic GLs. Glucobrassicanapin and sinigrin were two major individual GLs in Chinese kale baby-leaves, variation trends of which were consistent with the contents of total GLs and aliphatic GLs. From the heatmap analysis, and taking economic benefits into account, UVA-10 might be optimal for the production of high-quality Chinese kale baby-leaves in an artificial light plant factory.

**Keywords:** ultraviolet radiation; phytochemicals; glucosinolates; *Brassica alboglabra*





## 1. Introduction

Chinese kale (*Brassica alboglabra* Bailey) has been popularly consumed in South China and Southeast Asia owing to its important phytochemicals, including carotenoids, ascorbic acid, anthocyanin, polyphenols and glucosinolates [1]. Glucosinolates are the vital groups of health-promoting secondary metabolites in Brassicaceae vegetables. The hydrolysis of glucosinolates through myrosinase activity yields sulforaphane, which plays a beneficial role in the prevention and treatment of several diseases [2]. Many epidemiological studies indicated that diets rich in Brassica vegetables helped with reducing the risk of cancers and regulated immune and inflammatory responses [3,4]. Chinese kale baby-leaves are one of the forms in which *Brassicaceae* vegetables can be eaten. They are considered a "functional food" or "super-food" due to their higher contents of vitamins, minerals, and antioxidants compared to their mature counterparts. Thus, ready-to-eat baby-leaf vegetables have attracted great attentions in consumers and the food industry [5].

Light plays a vital role in driving photosynthetic activity and modifying the composition of plants. Different wavelengths and intensities of light have significant impacts on plant processes, such as photomorphogenesis and photoperiodism. Various photoreceptors of plants are sensitive to different light spectra, and then they activate downstream signaling transduction elements including some transcription factors which regulate many downstream genes, leading to physiological and biochemical changes in plants [6]. Rapid

advances in the development of LEDs provide a promising way to control the entire light spectrum and well-defined intensity under controlled conditions [7,8]. Moreover, the application of supplemental light has become a more economic and energetically efficient method in controlled environments, which is beneficial for the yield and quality of leafy vegetables [9].

Ultraviolet light (200–400 nm) takes up a small proportion of the solar spectrum, but plays a vital role in plant growth and development, which comprises up to 7% of the total sunlight-reaching terrestrial plants on the Earth's surface [10]. UV light is categorized into UV-C (200–280 nm), UV-B (280–320 nm), and UV-A (320–400 nm) spectra [11]. An enormous number of studies have highlighted the important role of UV-B as regulators of plant growth with the in-depth researches of UV-B photoreceptor UV RESISTANT LOCUS8 (UVR8) [12,13]. UV-A radiation is the major component of the UV radiation on Earth as it penetrates almost unaltered through the stratospheric ozone layer and the atmosphere, accounting for 95% of UV radiation at sea level [14]. Up to now, there has been a lack of comprehensive understanding of the impacts of UV-A on plant physiology and biochemistry. Previous studies have indicated that UVA plays a pivotal role in maximizing phytochemical accumulation and improving health-protective properties in vegetables under artificial light plant factories [15–17]. Light intensity had a significant impact on growth and biochemical characteristics; however, there is relatively little information about the effect of UV-A with different intensities on the primary and secondary metabolism in plants.

In the present work, we aim to explore the effects of supplementary UV-A LED (380 $\pm$10 nm) treatment 10 days before harvest on Chinese kale baby-leaves, in order to evaluate the effects of supplementary UV-A intensity on the growth and health-promoting compounds production in Chinese kale baby-leaves in an artificial light plant factory.

## 2. Materials and Methods

### 2.1. Plant Materials and Experimental Designs

The experiment was conducted in an artificial light plant factory at the College of Horticulture, South China Agricultural University (23.16° N, 113.36° E). The seeds of Chinese kale (*Brassica alboglabra* Bailey) were sowed in a sponge block with a modified quarter-strength Hoagland solution (the full-strength nutrient solution was made up of the following elements: 944 mg·L$^{-1}$ Ca(NO$_3$)$_2$·4H$_2$O; 404 mg·L$^{-1}$ KNO$_3$; 160 mg·L$^{-1}$ NH$_4$NO$_3$; 200 mg·L$^{-1}$ KH$_2$PO$_4$; 348 mg·L$^{-1}$ K$_2$SO$_4$; 492 mg·L$^{-1}$ MgSO$_4$·7H$_2$O, EC$\approx$1.2 mS·cm$^{-1}$ and pH $\approx$ 6.4). The photosynthetic photon flux density (PPFD) was 300 µmol·m$^{-2}$·s$^{-1}$ white LED, 10/14 h (light/dark). The environmental temperature (24 $\pm$ 2 °C), relative humidity (65–75%), and CO2 concentration (500 $\pm$ 100 µmol·mol$^{-1}$) in the plant factory were measured by a series of corresponding sensors during the entire growth stage (Kebai Hongye Technology Co., Ltd., Beijing, China).

Six-layer cultivation shelves, in which each layer was equally divided into four individual cultivation units, were placed in the plant factory equipped with adjustable red-white-blue-UVA-far red LEDs (380 $\pm$ 10 nm, Chenghui Equipment Co., Ltd., Guangzhou, China) as light sources. Moreover, each cultivation unit consisted of 3 transplanting boards (95 × 60 × 3 cm). After 14 days, uniform seedlings with three expanded true leaf were transplanted into a deep flow technique system with half-strength Hoagland nutrient solution, for a further 10 days, with 42 plants per transplanting board, which represented one replicate. For each lighting treatment, three replications were performed. Two days after transplanting under the basal light (Red: white LEDs = 2:3 at PPFD of 250 µmol·m$^{-2}$·s$^{-1}$), four supplemental UV-A treatments were set up in the same layer with four individual cultivation units (with one light treatment per unit): CK (basal light, non-UV-A treated), basal light + 5 W·m$^{-2}$ UV-A (UVA-5, nearly 10 µmol·m$^{-2}$·s$^{-1}$), basal light + 10 W·m$^{-2}$ UV-A (UVA-10, nearly 20 µmol·m$^{-2}$·s$^{-1}$), and basal light + 15 W·m$^{-2}$ UV-A (UVA-15, nearly 30 µmol·m$^{-2}$·s$^{-1}$), respectively. The photoperiod was set to 12 h (6:00–18:00) in all treatments by adjusting LED light panels. Except for changes in light environment

and transplanting density, all growth environments remained the same before and after transplanting. To prevent plants from being disturbed by different light environments, each cultivation unit was separated with a shading cloth. Light spectrograms measured by a spectroradiometer (ALP-01, Asensetek, Taiwan) were shown in Figure 1.

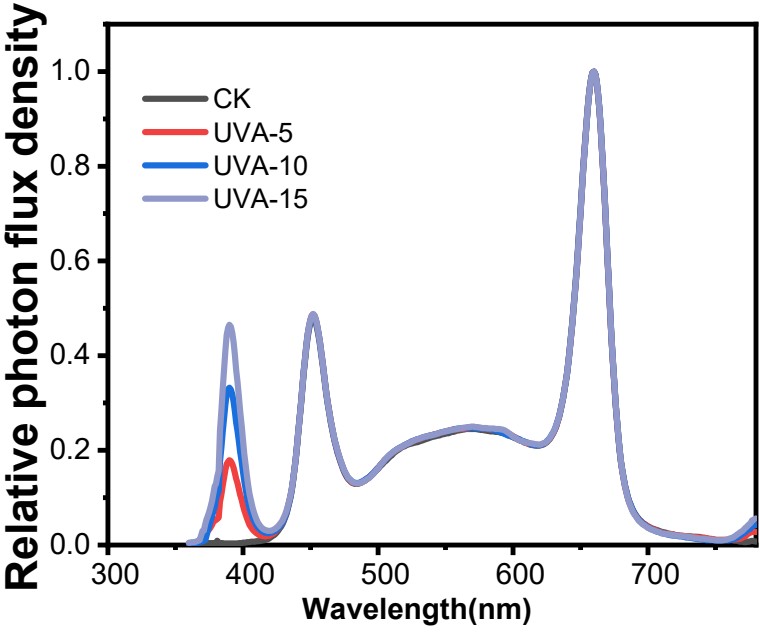

**Figure 1.** Spectral composition in the four treatments.

### 2.2. Agronomy Traits Measurements

Chinese kale baby-leaves plants were harvested from 10 plants per treatment at 10 days after UV-A supplementing (Figure 2). The internode length, stem diameter, and canopy diameter (the maximum width of plant shoot) of Chinese kale baby-leaves were measured by a rule. Fresh and dry weight (determined after 48 h at 75 °C in a drying oven) were weighed on an analytical balance. Plants sampled from different treatments were immediately frozen in liquid nitrogen and kept at −40 °C for analysis. Three analytical replications (6 plants per replication) were used in each biochemical measurement.

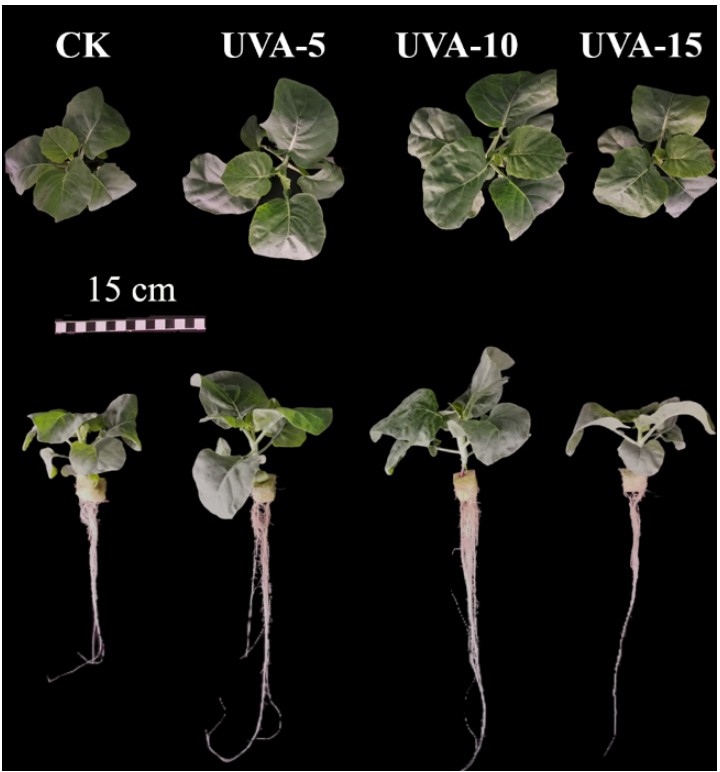

**Figure 2.** The Chinese kale baby-leaves morphology at 10 days after treatments. CK = control. UVA-5, UVA-10, and UVA-15 indicated UV-A intensity of 5, 10, 15 W·m$^{-2}$, respectively.

*2.3. Phytochemical Determinations*

Contents of photosynthetic pigment were extracted from fresh samples (0.5 g) with 25 mL acetone-alcohol mixture (acetone: alcohol = 1:1, *v/v*) until the leaf tissue turned white at 26 °C for 24 h in darkness [18]. Then, they were measured at 645, 663 and 440 nm by UV-spectrophotometer (Shimadzu UV-16A, Shimadzu, Corporation, Kyoto, Japan). The contents of chlorophyll a (Chl a), chlorophyll b (Chl b), chlorophyll (a + b) (Chl (a + b)) and carotenoid were quantified as follows:

$$\text{Chl a (mg·L}^{-1}) = 12.7 \times \text{OD663} - 2.69 \times \text{OD645}$$

$$\text{Chl b (mg·L}^{-1}) = 22.9 \times \text{OD645} - 4.86 \times \text{OD663}$$

$$\text{Chl (a + b) (mg·L}^{-1}) = 8.02 \times \text{OD663} + 20.20 \times \text{OD645}$$

$$\text{Carotenoids (mg·L}^{-1}) = 4.7 \times \text{OD440} - 0.27 \times \text{Chl (a + b)}$$

$$\text{Photosynthetic pigment (mg·g}^{-1}) = \text{Photosynthetic pigment (mg·L}^{-1}) \times 25 \text{ mL} \times 10^{-3}/0.5 \text{ g}$$

Soluble sugar content was determined using the sulfuric acid anthrone method [19]. Fresh sample powder (0.5 g) was added to 8 mL 80% ethanol and then on a water bath (80 °C) for 40 min after adding activated carbon powder (10 mg). The completed extract had a volume of 25 mL with 80% ethanol. The 0.2 mL supernatant, 0.8 mL distilled water and 5 mL sulfuric acid anthrone reagent were mixed and boiled for 10 min. Soluble sugar content was measured at 625 nm using a spectrophotometer.

Soluble protein content was measured according to Coomassie brilliant blue G-250 dye method [20]. Fresh sample powder (0.5 g) was mixed with 8 mL distilled water then centrifuged at 3000 rpm for 10 min at 4 °C. After that, 0.2 mL supernatant was combined with 0.8 mL distilled water and 5 mL Coomassie brilliant blue G-250 solution (0.1 g·L$^{-1}$). After 5 min, absorbance was read at 595 nm by a UV-spectrophotometer.

Nitrate content was determined as the method described by Cataldo [21]. Fresh samples (0.5 g) were mixed with 10 mL distilled water then boiled for 30 min. After filtering, 0.1 mL supernatant was added to 0.4 mL 5% salicylic and sulfuric acid and 9.5 mL 8% NaOH. The absorbance was measured at a wavelength of 410 nm.

The content of vitamin C was eluted from fresh samples (0.5 g) with 25 mL oxalic acid EDTA solution (200 Mm EDTA and 50 Mm oxalic acid) [22]. Next, 10 mL extracting solution was added to 900 1 mL 3% $HPO_3$, 2 mL 5% $H_2SO_4$ and 4 mL 5% $H_8MoN_2O_4$. After 15 min, the absorbance was measured at a wavelength of 705 nm.

The 2, 2-diphenyl-1-picrylhydrazyl (DPPH) radical scavenging rate was carried out based on Tadolini [23]. Fresh samples (0.5 g) were mixed with 8 mL ethanol, standing for 30 min in darkness, then the sample centrifuged at 3000 rpm for 15 min. An amount of 2 mL supernatant was mixed with 2.0 mL DPPH solution (0.0080 g DPPH in 100 mL ethanol). Absorbance of mixed solution was read at 517 nm.

The value of ferric-reducing antioxidant power (FRAP) was analyzed according to Benzie and Strain [24]. 0.4 mL sample solution (The extraction method of FRAP was the same as that of DPPH) was mixed with 3.6 mL mixed solution (0.3 mol·$L^{-1}$ acetate flavon; 10 mmol·$L^{-1}$ TPTZ; 20 mmol·$L^{-1}$ $FeCl_3$, 10:1:1, *v:v:v*) and incubated at 37 °C in water bath for 10 min. Absorbance was read at 593 nm.

Polyphenol content was measured based on the Folin–Ciocalteu assay [25]. The 1 mL supernatant (the extraction method of polyphenol was the same as that of DPPH) was mixed with 0.5 mL foline-phenol and 11.5 mL 26.7% sodium carbonate, then 7 mL distilled water was added into the mixed solution. After 2 h, absorbance was read at 510 nm by UV-spectrophotometer.

Measuring the total flavonoid contents was based on the method of Li [26]. The 1 mL extract solution (The extraction method of flavonoid was the same as that of DPPH) was added to 0.7 mL 5% $NaNO_2$. After 5 min, the reaction solution was mixed with 0.7 mL 10% $Al(NO_3)_3$, and then 6 min later, the mixture was added 5 mL 5% NaOH, and 10 min later, absorbance was read at 760 nm by UV-spectrophotometer.

The determination of glucosinolates was performed as earlier described by Qian [27] with minor modifications. The freeze-dried samples (approximately 0.2 g) were extracted for 20 min in a water bath at 75 °C with 4 mL 70% methanol. Then, they were homogenized with 0.4 mol·$L^{-1}$ barium acetate (2 mL) at 4000 rpm for 10 min. The above extraction steps were repeated once, the supernatants were combined, then they were loaded onto a mini column containing 500 mL DEAE-Sephadex A-25 that had been conditioned with 2 mol·$L^{-1}$ acetic acid and washed with 6 mol·$L^{-1}$ imidazole formate. After loading, the column was washed with 0.02 M sodium acetate buffer. Then, 500 mL sulfatase solution (Sigma-Aldrich, Steinheim, Germany) was added, and the preparation was incubated overnight. The latter was dissolved in 2 mL distilled water and filtered through a 0.22 μm membrane filter.

HPLC analyses were conducted on a Waters e2695 liquid chromatograph (Waters Crop., Milliford, MA, USA) with a reversed-phase C 18 column (5μm, 250 mm × 4.6 mm; Waters SunFire C18, Waters, Milford, MA 01757, USA). The column was maintained at 30 °C and 20 μL injection volume. A binary gradient was used: 0–32 min 0–20% eluent A; 32–38 min 20% A; 39–40 min 20–100% A. The eluents were (A) acetonitrile and (B) distilled water, with a flow rate of 1 mL·$min^{-1}$. The detection wavelength was recorded at 229 nm. Contents of the individual glucosinolate compounds were identified by their retention times and spectral data as compared by standards.

### 2.4. Statistical Analysis

All the assays were performed in triplicates. Significant differences among the treatments were determined by a one-way analysis of variance (ANOVA), using SPSS 22.0 software (SPSS Inc., Chicago, IL, USA). Significance at $p < 0.05$ was performed by Duncan's test. Heatmap synthesized analysis using TBtools software [28].

## 3. Results

### 3.1. Effect of Different Supplemental UV-A Intensity on The Morphology and Biomass of Chinese Kale Baby-Leaves

The morphology and biomass of Chinese kale baby-leaves were obviously affected by different supplemental UV-A intensities (Figure 3). Compared to CK, the plant fresh weight and dry weight of shoot were not affected under UVA-5, while they significantly increased by 42% and 41% under UVA-10, and decreased by 20% and 20% under UVA-15 treatment. The plant fresh weight and dry weight of root showed a similar pattern to shoot fresh weight and dry weight were in response to different supplemental UV-A intensities. The internode length of Chinese kale baby-leaves under UVA-5 and UVA-10 increased by 6% and 30%, respectively, but decreased by 5% under UVA-15. Similarly, the stem diameter under UVA-5 and UVA-10 increased by 5% and 13%, respectively, but decreased by 20% under UVA-15. There was no striking difference in canopy diameter of Chinese kale baby-leaves under UVA-15, compared with CK, while this increased by 9% and 22% under UVA-5 and UVA-10, respectively.

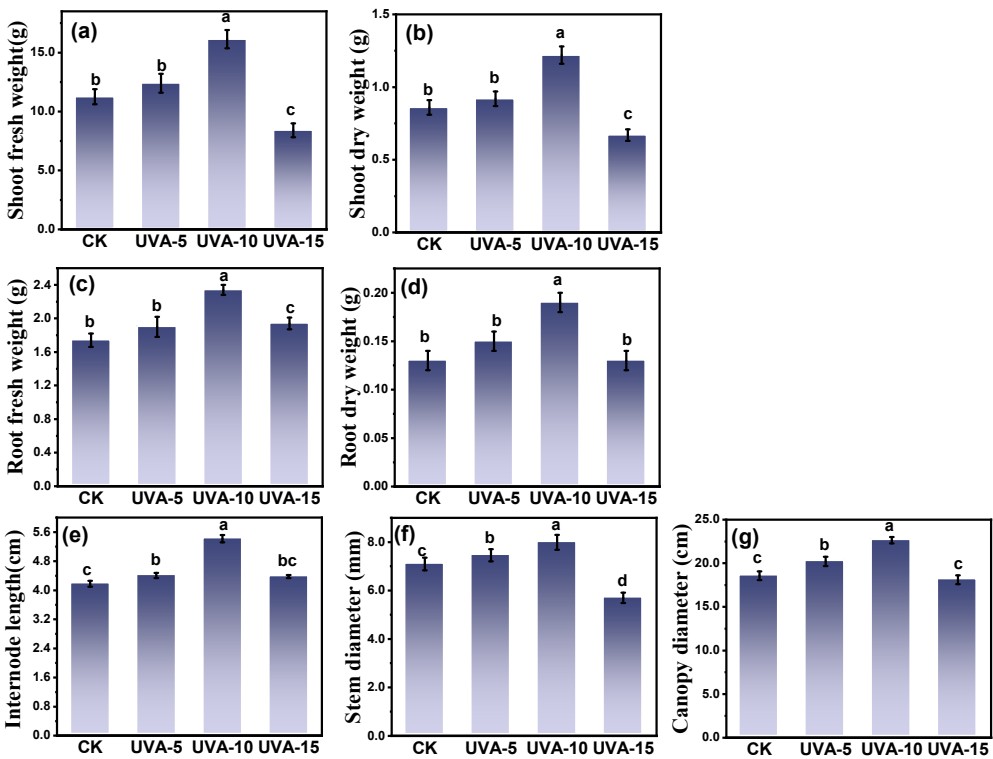

**Figure 3.** Agronomy traits of Chinese kale baby-leaves under different supplemental UV-A intensities. (**a**) Shoot fresh weight, (**b**) Shoot dry weight, (**c**) Root fresh weight, (**d**) Root dry weight, (**e**) Internode length, (**f**) Stem diameter, (**g**) Canopy diameter of Chinese kale baby-leaves. All data are presented as mean ± standard error (*n* = 10). Different letters (**a**–**d**) marked upon the bar plots indicated significant differences between treatments by Duncan's multiple (*p* < 0.05).

### 3.2. Effect of Different Supplemental UV-A Intensity on the Photosynthetic Pigment Contents of Chinese Kale Baby-Leaves

Different influences of supplemental UV-A intensities on the photosynthetic pigment contents of Chinese kale baby-leaves are presented in Table 1. In UVA-5 treatment, the contents of Chl a and Chl (a + b) increased by 3% and 8%, respectively, while the ratio of Chl a/Chl b decreased by 4%, and chlorophyll/carotenoids ratio was not affected. Expect for the fact that Chl a/Chl b ratio decreased by 4%, there were no significant differences between the contents of chlorophyll and carotenoids under UVA-10. The contents of Chl a,

Chl (a + b) and chlorophyll/carotenoids ratio in UVA-15 decreased by 12%, 9% and 10%, respectively.

**Table 1.** Photosynthesis pigments of Chinese kale baby-leaves under different supplemental UV-A intensity.

| Treatments | Chlorophyll Content (mg/g) | | | Carotenoid (mg/g) | Chl a/Chl b | Chlorophyll /Carotenoid |
|---|---|---|---|---|---|---|
| | Chl a | Chl b | Chl (a + b) | | | |
| CK | 1.16 ± 0.04 ab | 0.54 ± 0.08 a | 1.49 ± 0.07 ab | 0.21 ± 0.01 a | 1.86 ± 0.02 a | 7.64 ± 0.35 ab |
| UVA-5 | 1.19 ± 0.06 a | 0.67 ± 0.04 a | 1.61 ± 0.09 a | 0.20 ± 0.00 a | 1.78 ± 0.02 b | 8.38 ± 0.56 a |
| UVA-10 | 1.11 ± 0.04 ab | 0.61 ± 0.03 a | 1.49 ± 0.06 ab | 0.20 ± 0.00 a | 1.79 ± 0.02 b | 7.54 ± 0.32 ab |
| UVA-15 | 1.02 ± 0.02 b | 0.54 ± 0.02 a | 1.36 ± 0.03 b | 0.21 ± 0.00 a | 1.87 ± 0.01 a | 6.84 ± 0.11 b |

UVA-5, UVA-10 and UVA-15 indicated UV-A intensity of 5, 10, 15 $W\cdot m^{-2}$, respectively. The results presented were means ± standard error ($n$ = 3). Different letters marked in tables indicate significant differences between treatments by Duncan's multiple ($p < 0.05$).

### 3.3. Effect of Different Supplemental UV-A Intensity on Nutritional Compounds of Chinese Kale Baby-Leaves

The contents of nutritional compounds in Chinese kale baby-leaves were significantly affected by supplemental UV-A intensities (Figure 4). Contents of soluble sugars decreased by 4% under UVA-5, but they were not affected under UVA-10 and UVA-15, compared with CK (Figure 4a). The content of soluble protein under UVA-5 and UVA-10 decreased by 4% and 7%, respectively, while a significant increase was found in UVA-15 (Figure 4b). The Vc contents decreased under UVA-5 compared with CK, but no significant differences were found under UVA-10 and UVA-15 (Figure 4c). The nitrate contents increased by 23% under UVA-5, while there were no striking changes under UVA-10 and UVA-15 (Figure 4d).

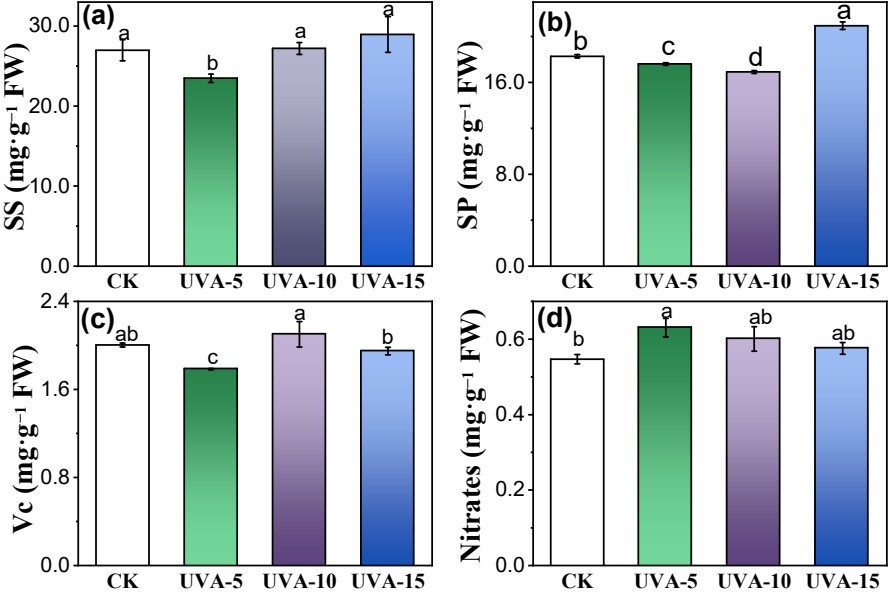

**Figure 4.** Nutritional compounds of Chinese kale baby-leaves under different supplemental UV-A intensity. (**a**) SS = soluble sugars, (**b**) SP = soluble protein, (**c**) Vc = vitamin C, (**d**) nitrate. The results presented were means± standard error ($n$ = 3). Different letters (**a–d**) on the bar plots indicate significant difference at $p < 0.05$ using one-way analysis of variance with Duncan's multiple-range test.

### 3.4. Effect of Different Supplemental UV-A Intensity on Antioxidant Content and Capacity of Chinese Kale Baby-Leaves

Antioxidant content and capacity in Chinese kale baby-leaves were significantly relevant to supplemental UV-A intensity (Figure 5). No obvious difference was observed in the DPPH of Chinese kale baby-leaves between UV-A and CK (Figure 5a). FRAP in Chinese

kale baby-leaves markedly increased under UVA-10 (Figure 5b), while there were no significant differences among CK, UVA-5 and UVA-15. The highest total polyphenol content was found in UVA-10, which was 19% higher than CK, and in UVA-5 and UVA-15, total polyphenol contents were 10% and 4% higher than CK, respectively (Figure 5d). Total flavonoid contents significantly increased under supplemental UV-A; those under UVA-10 and UVA-15 were significantly higher than CK by 64 and 23%, respectively, though no significant increase was found in UVA-5 (Figure 5c).

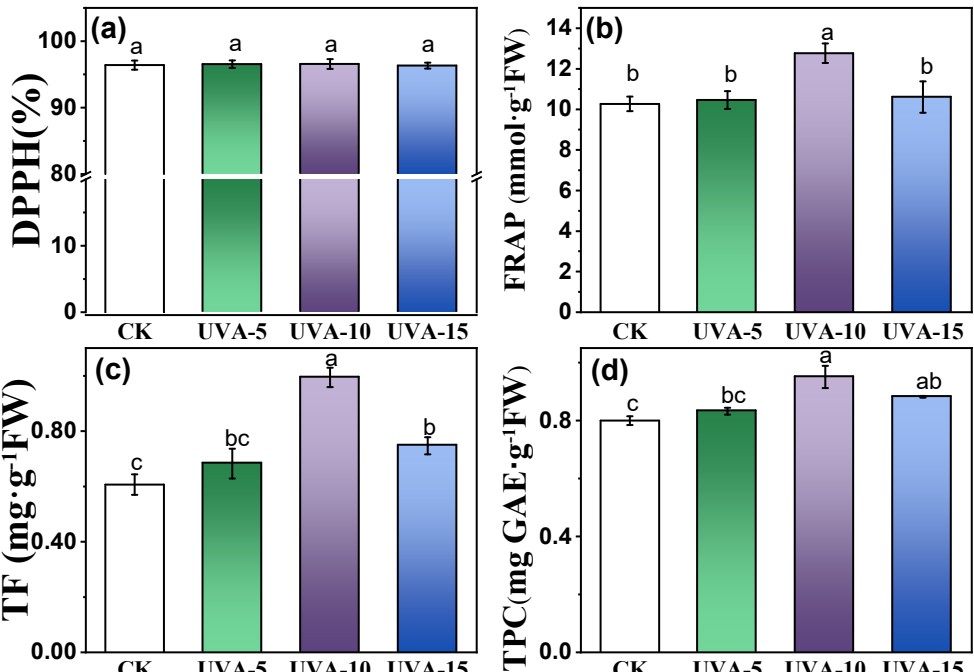

**Figure 5.** Antioxidant content and capacity of Chinese kale baby-leaves under different supplemental UV-A intensity. (**a**) DPPH = 2,2-diphenyl-1-picrylhydrazyl, (**b**) FRAP = ferric-reducing antioxidant power, (**c**) TF = total flavonoid, (**d**) TPC = total phenolics. The results presented were means $\pm$ standard error ($n$ = 3). Different letters (**a**–**d**) upon the bar plots indicate significant difference at $p < 0.05$ using one-way analysis of variance with Duncan's multiple-range test.

### 3.5. Effect of Different Supplemental UV-A Intensity on Glucosinolate Composition and Content of Chinese Kale Baby-Leaves

In this study, eight different glucosinolates (GLs) were identified in Chinese kale baby-leaves by HPLC based on their retention times (Figure 6a), which included four aliphatic GLs (progoitrin(PRO), sinigrin(SIN), glucoraphanin(GRA), and glucobrassicanapin (GBN)), and four indol GLs (4-hydroxy glucobrassicin(4-HGBS), glucobrassicin(GBS), 4-methoxy glucobrassicin(4-MGBS) and neoglucobrassicin(NGBS)). Aliphatic GLs contents as the most abundant GLs found in Chinese kale baby-leaves accounted for about 80% of the total GLs, which significantly increased with an increase in supplemental UV-A intensity (Figure 6). Compared with CK, the total aliphatic GLs contents under UVA-5, UVA-10 and UVA-15 were significantly higher by 19, 38 and 57%, respectively. Although PRO under UVA-10 did not differ significantly from CK, as well as PRO and GRA under UVA-5, contents of PRO, SIN and GRA showed an increasing trend with an increase in supplemental UV-A intensity; those under UVA-15, UVA-10 and UVA-5 were higher than CK by 78–93%, 15–72% and 18–51%, respectively. The total indolic GLs contents under supplemental UV-A treatments exhibited no striking change, except for a significant increase was detected only in the 4-HGBS content. Therefore, the increase in total GLs content under supplemental UV-A was due to UV-A causing a significant increase in PRO, SIN and GRA, which were the predominant GLs in Chinese kale baby-leaves. Total GLs content in Chinese kale baby-leaves increased with an increase in supplemental UV-A intensity.

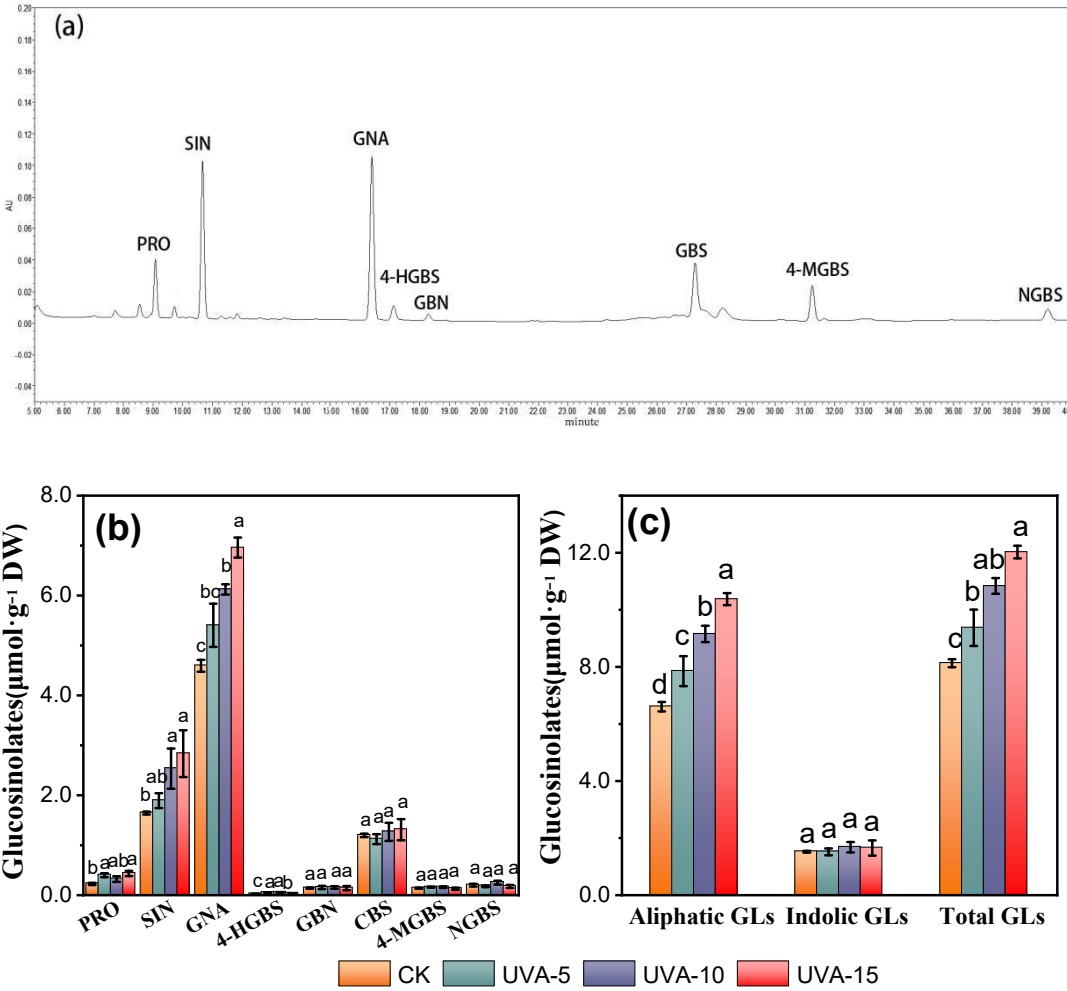

**Figure 6.** Glucosinolate composition and contents of Chinese kale baby-leaves under different supplemental UV-A intensities. The results presented were means ± standard error (*n* = 3). Different letters (**a–c**) upon the bar plots indicate significant difference at *p* < 0.05 using one-way analysis of variance with Duncan's multiple-range test.

### 3.6. Heatmap Analysis of Functional and Nutritional Aspects of Chinese Kale Baby-Leaves under Different Supplemental UVA Light Intensity

A heatmap synthesized the responses of yield, phytochemicals, and GLs to supplemental UV-A, providing an integrated view about agronomy traits, nutritional and functional profile of Chinese kale baby-leaves in response to UV-A intensity (Figure 7). The parameters could be grouped into two main clusters by the hierarchical clustering analysis, corresponding to the light treatments, in which UVA-5, UVA-15 and CK clusters were the closest to each other in the measured parameter responses. The UVA-10 cluster was obviously distinguished from other clusters by its higher biomass (fresh weight, dry weight, inter-node length, stem diameter, canopy diameter) and antioxidant compounds and activities (Vitamin C, phenolics, flavonoid, FRAP). The analysis of the heatmap indicated that UVA-10 played an important role in promoting plant growth and elevating the accumulation of functional phytochemicals of Chinese kale baby-leaves.

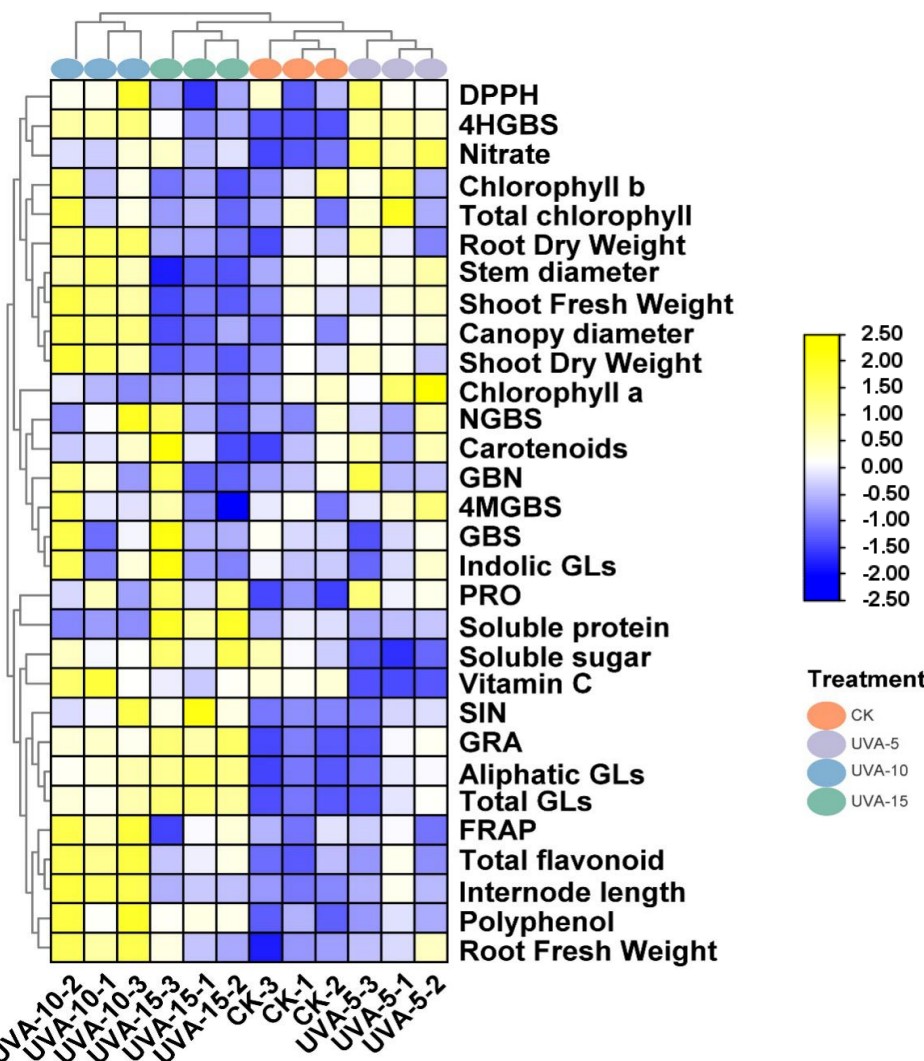

**Figure 7.** Heatmap analysis summarized the effect of different supplemental UV-A intensity on growth and quality of Chinese kale baby-leaves. Results were visualized using a false color scale, with yellow and purple indicating an increase and decrease, respectively, in the response parameters.

## 4. Discussion

Supplementary UV-A intensity notably affected plant growth in this study. The plant biomass of Chinese kale baby-leaves under UVA-10 was higher than other UV-A treatments (Figure 3), while they showed a lower biomass under UVA-15 than CK. This indicates that a lower supplemental UV-A intensity (UVA-10) increased the biomass of Chinese kale baby-leaves, while a higher UV-A intensity (UVA-15) inhibited biomass accumulation (Figure 3), and UVA-10 was more favorable for Chinese kale baby-leaves plant growth in an artificial light plant factory (Figure 3). Studies have shown that UV-A exerted a stimulatory effect on biomass in radish [29], lettuce [30], and six Mediterranean plant species [31], while an inhibitory effect was found in others, such as in lettuce [32], cucumber [33], as well as in some C3 and C4 plants [34]. Different wavebands of UV-A LEDs (370 nm or 385 nm) with 30 W·m$^{-2}$ for 5 days induced significant increases in the biomass of kale, and the fresh and dry weights of shoots and roots, leaf area, and specific leaf weight were significantly higher than in control plants [15]. Exposing Chinese kale and pak-choi baby-leaves to 40 W·m$^{-2}$ UV-A (380 nm) resulted in higher biomass than CK, with a significant increase in plant height, leaf length, and leaf numbers [35]. UV-A (365 nm) irradiation at 10, 20, and 30 μmol·m$^{-2}$·s$^{-1}$ significantly stimulated biomass production of indoor cultivated lettuce, yielding 15–29% higher shoot dry weight, and

the stimulating effect of UV-A on lettuce growth was not a simple linear dose–response relationship [17]. Lettuce might have a saturation response to UV-A intensity. In this study, Chinese kale baby-leaves morphological responses were dependent on the received UV-A dosage (Figure 3). The increase in biomass of Chinese kale baby-leaves exposed to lower UV-A intensity might be attributed to UV-A as one of the driving forces behind plant photosynthesis, which helped to elevate stomatal opening and $CO_2$ absorbing [36], and consequently resulted in the growth of adaxial epidermal cells and an increase in leaf area. Moreover, UVA photoreceptors (cryptochromes) play an important role in leaf photosynthetic development since they could regulate the expression of chloroplast genes which associated with the transcription and expression of photosystem II (PSII) encoding genes [37,38]. However, excessive UV-A levels might cause damage to the PSII protein complex and reduce quantum efficiency, which was similar to UV-B [39].

An increase in plant biomass under UV-A irradiation might be closely linked to higher photosynthetic pigments contents and photosynthetic activity. The chlorophyll contents increased rapidly in *Hippophae rhamnoides* [40], and canola (*Brassica napus*) [41] under UV-A radiation, but decreased in lettuce [42] and some C3 and C4 plants (cotton, wheat, amaranthus, sorghum) [34]. Supplemental UVA (368 nm) of 0.45 W·m$^{-2}$ for 15 days in "Roma" tomato seedlings led to a significant stimulation of photosynthetic pigments [43]. Some studies showed that exposure to UV-A radiation increased the contents of chlorophylls and carotenoids, thereby promoting plant growth [44,45]. Chl a/Chl b reflects the adaptability of plants to light intensity. Lower Chl a/Chl b indicated that plants had a stronger ability to the utilization of weak light [46]. In this study, the contents of chlorophyll increased under UVA-5 but decreased under UVA-10 and UVA-15 (Table 1). Moreover, a lower ratio of Chl a/Chl b was found in UVA-10 (Table 1), which indicated that chlorophylls and carotenoids of Chinese kale baby-leaves might directly absorb UVA light as light energy for photosynthesis, enhancing photosynthesis and promoting plant growth [47,48]. Moreover, UV-induced violet-blue-green fluorescence could be captured by photosynthetic pigments to enhance electron transport; hence, the photosynthetic rates were improved under UV-A supplementation to the non-saturating levels of visible light [49]. Inversely, energy-rich UV radiation results in the generation of free radicals which are a damaging factor for the photosynthetic machinery, including chloroplasts and the degradation of photosynthetic pigments [47]. Moreover, UV-A could cause damage to the photosystem II protein complex and decrease quantum efficiency [47]. UVA-15 led to the reduction in Chl a and chlorophyll/carotenoid ratio, which might be an optical photoprotective mechanism against damage of photosynthesis induced by UV irradiation [48,50]. UV-A doses used in Chinese kale baby-leaves led to different effects on chlorophyll contents (Table 1), indicating that the UV-A dose effect should not be ignored. The lower production of Chinese kale baby-leaves biomass might be the result of a dose–response of UV-A [43].

Soluble sugars, soluble proteins, nitrate and Vc fulfill very important roles in evaluating the nutritional quality of vegetables. UV-A supplementation could lead to the remarkable variation in phytochemicals such as sugar and protein by regulating the processes of carbon and nitrogen metabolism and photosynthesis of vegetables [51]. Supplemental UV-A resulted in 6.4 times higher nitrate contents in green leaf lettuce and 62% higher nitrate contents in red leaf lettuce, compared to the blank group [52]. A similar result was found in our study; that the nitrate contents of Chinese kale baby-leaves slightly increased by supplemental UV-A radiation (Figure 4d), which might due to UV-A could affect the gene expression and enzyme activity related to nitrogen metabolism in plants [53]. Moreover, glutamine contents were significantly 2.8 times higher than control under supplemental UV-A in green leaf lettuce, but the free amino acid contents in red leaf lettuce decreased [52]. Furthermore, the highest soluble protein content of pak-choi was found in UV-A (380 nm) treatment, while no obvious changes in free amino acids contents in red-leaf pak-choi were found between UV-A and CK [54]. Nitrate reductase activities, contents of soluble protein and Vc in radish plants were promoted by the UV-A (about 305–400 nm) radiation [29]. In this study, contents of total protein, soluble sugar and Vc contents respond to vary-

ing intensities of UV-A, which showed a significant reduction under UV-5 (Figure 4a–c). However, the opposite effect was detected in UV-15, with an obvious increase in total protein contents (Figure 4b). Our results were inconsistent with results showing that soluble sugar and protein contents in lettuce increased under UV-A supplementation of 10, 20 and 30 $\mu mol \cdot m^{-2} \cdot s^{-1}$ [17]. A decrease in soluble sugar and protein content was found in UV-A treatment, which might indicate the normal consumption of these reserve carbohydrates; once these baby-leaves increased biomass (leaf area) and the photosynthetic pigments did not change, the photosynthate might be used as the energy source for plant growth [43]. Phloem integrity might be disrupted by UV-A radiation, which might exert an impact on carbohydrate metabolism in plants [55]. The effect of supplemental UV-A on the phytochemicals of Chinese kale baby-leaves in an artificial light plant factory exhibited a different response to the UV-A dose.

DPPH free radical scavenging rate and ferric-reducing antioxidant power (FRAP) are often used to assess the total antioxidant capacity of vegetables. The contents of total phenols and flavonoids were predominant health-promoting compounds which exert a crucial impact on the plant defense system and maintain many biological activities in the human body [56]. UV radiation is considered to be an effective mutagenic agent which can promote the production of diverse secondary metabolites such as flavonoids, phenols, alkaloids, and terpenoids in plants [57,58]. Compared with CK, the DPPH and FRAP of red-leaf and green-leaf pak-choi pronouncedly increased by 35.39% and 33.32%, respectively, under the UV-A treatment. Additionally, UV-A showed a promoting effect on the contents of TPC, TF, and TA in two pak-choi cultivars [54]. Additionally, broccoli sprouts exposed to 3.16, 4.05 $W \cdot m^{-2}$ UV-A for 120 min were beneficial for the production of sinapic acid, gallic acid and gallic acid derivatives [58]. The expressions of structural genes regulating phenols and flavonoids biosynthesis in plants, such as *PAL*, *CHS*, *DFR* and *AN* could be tremendously up-regulated by UV-A radiation [59–61]. No significant difference or slight increase was found in DPPH scavenging activity, FRAP and flavonoids and polyphenols contents of Chinese kale baby-leaves under UVA-5 and UVA-15 compared with CK, but UVA-10 could significantly improve the antioxidant activity and antioxidant content of Chinese kale baby-leaves (Figure 5). Therefore, the stimulating effect of UV-A on the antioxidant of Chinese kale baby-leaves depended on the dose of UV-A illumination.

Glucosinolates are a class of sulfur-rich secondary metabolites widely found in Brassica vegetables, which have received considerable attention recently due to the anticarcinogenic properties of their hydrolysis product, sulforaphane [62]. With increasing supplementary UV-A intensity, the contents of aliphatic GLs remarkably increased in Chinese kale baby-leaves (Figure 6b,c). No significant differences were found in the indol GLs contents in Chinese kale baby-leaves among the treatments, except for 4-HGBS, which significantly increased under supplementary UV-A (Figure 6b). In broccoli sprouts, UV-A (3.16 $W \cdot m^{-2}$ for 120 min) radiation showed higher accumulation of both aliphatic and indolic GLs in sprouts harvested 24 h afterwards, compared to CK [58]. However, all individual GLs contents did not show any change after UV-A treatment (only GER was significantly decreased by 27.6%) in 7-day-old broccoli sprouts under UV-A (9.47 $W \cdot m^{-2}$) radiation for 120 min [63]. UV-A significantly decreased progoitrin content, while 400 nm UV-A had a promotion effect in increasing the contents of sinigrin and glucobrassicin in Chinese kale [35]. Meanwhile, 400 nm UV-A was able to increase the contents of glucoraphanin, sinigrin, and glucobrassicin in pak-choi [35]. Therefore, the biosynthesis of glucosinolates responding to UV-A varied according to the duration and dose of exposure. Aliphatic GLs were the dominant GLs in Chinese kale baby-leaves, constituting approximately 80% of the total GLs contents (Figure 6). UV-A radiation could stimulate the biosynthesis and accumulation of GLs in Chinese kale, which might be caused by the activation of transcription factor genes such as HY5, after the UV-A photoreceptor senses the signal. HY5 could regulate the transcription levels of sulfur assimilation genes *ATPs*, *APK* and *APR*, and then affect the expression of glucosinolates biosynthesis genes, e.g., *SOT16*, *SOT17*, *SOT18*, *MAM-L*, *CYP79F1* and *CYP79B2* [64]. From the results of this study, GLs con-

tents significantly increased under UV-A might attribute to that UV-A could up-regulate the gene expressions of the key enzymes related to glucosinolates metabolites [65,66]. It could increase the expression of transcription factors (e.g., *DOF1.1, MYB41, MYB28, MYB34*) and cure structure genes of glucosinolates which involve *BCATs, MAMs, CYP79s, CYP83, AOPs* and other gene families. UV-A could increase the glucosinolates contents in Chinese kale baby-leaves, the difference in the underlying mechanism still needs further research.

## 5. Conclusions

The response of Chinese kale baby-leaves growth and metabolism highly depended on UV-A intensity. Our results indicate that UVA-10 was beneficial for Chinese kale baby-leaves' plant biomass and morphology, while UVA-15 inhibited the growth of Chinese kale baby-leaves. A significant increasing trend for antioxidant compounds was found under the UVA-10 treatment. The glucosinolates contents in Chinese kale baby-leaves significantly increased with the increase in UV-A intensity, and the highest contents were found under UVA-15. Taking economic benefits into account, UVA-10 (10 $W \cdot m^{-2}$ applied 10 days before harvest) might be optimal for the production of high-quality Chinese kale baby-leaves in artificial light plant factory. Further studies are needed to explore how UV-A regulates phytochemicals through light signal transduction pathways.

**Author Contributions:** Conceptualization, R.H. and H.L.; methodology, R.H., M.G. and Y.L.; software, R.H.; validation, W.S., Y.Z., S.S. and H.L.; investigation, W.S.; resources, H.L.; data curation, R He., M.G. and H.L.; writing—original draft preparation, R H. and H.L.; writing—review and editing, R.H. and H.L. All authors have read and agreed to the published version of the manuscript.

**Funding:** This work was supported by grant from the Key-Area Research and Development Program of Guangdong Province (2019B020214005, 2019B020222003).

**Institutional Review Board Statement:** Not applicable.

**Informed Consent Statement:** Not applicable.

**Data Availability Statement:** Not applicable.

**Conflicts of Interest:** The authors declare that they have no known competing financial interests or personal relationships that could have appeared to influence the work reported in this paper.

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
