# Peer review of "Supplemental UV-A Affects Growth and Antioxidants of Chinese Kale Baby-Leaves in Artificial Light Plant Factory"

_horticulturae, doi:10.3390/horticulturae7090294_

Round 1
Reviewer 1 Report
please find attached file

Reviewer 2 Report
The article deals with the quantification of the effect of supplemental UV-A on growth and antioxidiants production in the case of Chinese kale Baby-leaves. This study contributes to the database concerning the effect of light quality on plant growth and development. Although the results the explication of the effects are given on the base on the literature or speculation, the study is revlevant for Horticulturae journal.
General comments
The article is very dense with a lots of results but is easy to read. In some cases I was wondering if it was necessary to have two decimal places after the decimal point for the percentage. I have only few specific comments concerning the discussion that can be improved.
Specific comments
There are two small comments :
L303 : “Short exposure”. This point is a real issue concerning the analysis of light quality effect. Could you discuss this point and give some elements on time application ?
L305 : when you write “growth” maybe it would be more clear to specify if it is growth in biomass or in dimensions.
L309 : µmol.m-2.s-1 is used here, maybe it would be interesting to use the same unit with your device to achieve the comparison.
L315 : what about the cryptochrome photoreceptor ?
L324 : see my previous comment concerning the unit J.m-2.s-1
L330 : are the plants stressed in your study ? What is the ratio ?
L371 : this paragraph could be imporved by giving some explanations.
L417 : Instead of speculating it is better to give some references or to give your own experimental results.
Reviewer 3 Report
The manuscript presents an experiment investigating the impact of supplemental UV-A light on plant growth and antioxidants of Chinese kale. The subject of the manuscript is interesting, but some additional detail is needed to more clearly understand the experimental methodology.
Specific comments:
Materials and Methods
- L76-77: Could you describe how the growing environment (PPFD, air temperature, and relative humidity) was monitored in the plant factory?
- L78-81: It is less clear how the hydroponic system was set up. What types of hydroponic system did you use? Was it deep water culture or NFT? What does the “plate” mean? Does it indicate “floating raft”?
- L78-81: Was the growing environment same before and after the transplanting? How did you ensure that plants are growing under the same environmental conditions under each lighting treatment?
- L81-87: It is less clear how the supplemental UV-A light was applied? Did you apply the basal light + UV-A light at the same time from 6 am to 6 pm?
- L85: Do you have any specific reason to use the unit of W∙m-2 for the intensity of UV-A light? As you used for the PPFD, photon flux density (in unit of µmol∙m–2∙s–1) is commonly used to quantify the light intensity for horticultural lighting. If possible, please use photon flux density (in unit of µmol∙m–2∙s–1) in the manuscript.
- L86-87: It is less clear how the lighting treatments were set up and monitored. Could you describe how you ensure that plants receive 250 µmol∙m–2∙s–1 of PPFD under each lighting treatment and 5, 10, and 15 W∙m-2 of UV-A?
- L89-95&174-179: How many plants did you use for data collection per each replication?
Figures and Tables
- Figures 3-6 and Table 1: What does the error bar indicate? Is it for the standard error or standard deviation for three replications (n=3)?
- Figure 6: Please indicate what (which lighting treatment) the color of each bar graph stands for.
